# HyperDisGAN: A Controllable Variety Generative Model Via Hyperplane Distances for Downstream Classifications

## Abstract

Despite the potential benefits of data augmentation for mitigating the data insufficiency, traditional augmentation methods primarily rely on the prior intra-domain knowledge. On the other hand, advanced generative adversarial networks (GANs) generate cross-domain samples with limited variety, particularly in small-scale datasets. In light of these challenges, we propose that accurately controlling the variation degrees of generated samples can reshape the decision boundary in the hyperplane space for the downstream classifications. To achieve this, we develop a novel *hyperplane distances GAN (HyperDisGAN)* that effectively controls the locations of generated cross-domain and intra-domain samples. The locations are respectively defined using the vertical distances of the cross-domain target samples to the optimal hyperplane and the horizontal distances of the intra-domain target samples to the source samples, which are determined by *Hinge Loss* and *Pythagorean Theorem*. Experimental results show that the proposed HyperDisGAN consistently yields significant improvements in terms of the accuracy (ACC) and the area under the receiver operating characteristic curve (AUC) on two small-scale natural and two medical datasets, in the hyperplane spaces of eleven downstream classification architectures. Our codes are available in the anonymous link: https://anonymous.4open.science/r/HyperDisGAN-ICLR2024.

## 1 Introduction

Deep neural networks achieve excellent performance in the computer vision fields (Alzubaidi et al., 2021), such as image classification (Krizhevsky et al., 2012), object detection (Redmon et al., 2016), image segmentation (Ronneberger et al., 2015) and image registration (Balakrishnan et al., 2019). In all these fields, a large-scale dataset containing sufficient supervisory information is crucial for effectively training of neural networks. However, in many realistic scenarios, neural networks can only be trained on the small-scale datasets, resulting in overfitting on the training set and poor generalization on the testing set. Although regularization techniques, such as parameter norm penalties, dropout (Srivastava et al., 2014), batch normalization (Ioffe & Szegedy, 2015), layer normalization (Ba et al., 2016) and group normalization (Wu & He, 2018), have been developed to prevent overfitting, data augmentation is another way addressing the training samples insufficiency.

Traditional augmentation method transforms the training samples by exploring prior intra-domain knowledge (Krizhevsky et al., 2012; Ciresan et al., 2011), including random cropping, rotations, etc. However, these methods are designed based on the specific scene, and the transformed samples have limited contributions to reshaping the decision boundaries for downstream classification tasks. Generative adversarial network (GAN) (Goodfellow et al., 2014) aims to generate the cross-domain samples having the same distribution with the target domain's samples. However, due to the issue of mode collapsing in GANs (Saxena & Cao, 2022), the generated samples' quality become uncertain and their variety is inferior to that of the real samples. Additionally, there is uncertainty regarding the domain labels of generated cross-domain samples. Briefly, the generated samples using above two main methods cannot guarantee their usefulness for the downstream classification tasks.

Previous augmentation methods blindly augmenting the training samples and make limited contributions to describing the decision boundary. This naturally raises a question, as illustrated in

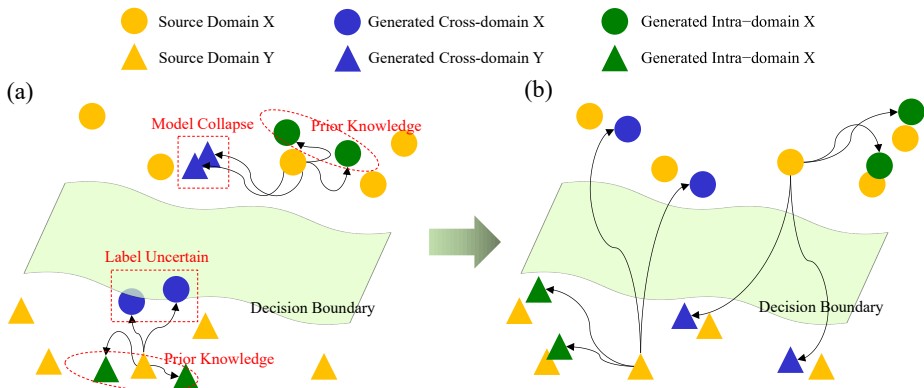

Figure 1: The illustration of data augmentation under two settings: (a) Not controlling the variation degrees. The intra-domain generated samples rely on the prior knowledge and make limited contributions to the decision boundary; The cross-domain generated samples degenerate to be intra-domain due to model collapse and have uncertain domain labels. (b) Controlling the variation degrees. All the generated samples are capable of reshaping the decision boundary in the hyper-space.

Figure 1: *If the variation degrees of generated samples are controlled, will a more accurate decision boundary be formed in the hyper-space for downstream classifications*? To answer this question, we conduct a comprehensive study on the *hyperplane distance GAN (HyperDisGAN)* controlling the locations of the generated samples adapting to the various hyperplane spaces of eleven classification architectures. Firstly, an auxiliary pre-trained classifier constructs a hyperplane space implying location information (including vertical distance and horizontal distance) by *Hinge Loss* (Rosasco et al., 2004). Secondly, the vertical distances from the cross-domain target samples to the optimal hyperplane and the horizontal distances from the intra-domain target samples to the source samples are taken as the controllable location parameters. Finally, a more precise hyperplane is constructed using the generated samples with controllable variations.

Our main contribution is three-fold: (1) A pre-trained hyperplane space based on hinge loss providing the location information of the real samples. (2) A novel HyperDisGAN controlling the locations of generated samples in the vertical and horizontal direction. (3) An effective data augmentation manner enabling reshaping the hyperplanes for various classifiers' architectures.

## 2 RELATED WORK

**Two-domain Image Transformation.** Recent works have achieved success in two-domain transformation. For instance, Pix2pix (Isola et al., 2017) learns the general image transformations in a supervised setting via L1+cGAN loss. However, it requires aligned image pairs due to the pixel-level reconstruction constraints. To alleviate the requirement of paired image supervision, unpaired two-domain transformation networks have been proposed. UNIT (Liu et al., 2017) is a coupled VAE-GAN algorithm based on a shared-latent space assumption. CycleGAN (Zhu et al., 2017) and DiscoGAN (Kim et al., 2017) enforce bidirectional transformations by utilizing a cycle consistency loss. In this study, the proposed HyperDisGAN enables not only the unpaired cross-domain image transformation but also the unpaired intra-domain transformation. The baseline CycleGAN enabling unpaired cross-domain transformation is presented in Appendix A.

**GAN-based Augmentation Schemes.** Recently, GAN (Goodfellow et al., 2014) and its variations have been employed as the augmentation tools for the image classification. For instance, DCGAN (Radford et al., 2016) generates high-quality CT images for each liver lesion class (Frid-Adar et al., 2018). In this augmentation scheme, the generated inner-domain samples cannot well describe the decision boundary. Auxiliary classifiers have been combined with GANs to create soft labels for generated cross-domain data (Shi et al., 2018; Haque, 2021). They alleviate the generated samples' authenticity by setting weights for them. However, soft labels are not precise to benefit the downstream classification tasks, and finding appropriate classification weights for unreliable data is difficult. Other schemes generate cross-domain data via CycleGAN (Chen et al., 2020; Cap et al.,

2022; Bargshady et al., 2022). Unfortunately, the generated samples close to the distribution of the source domain are given the target-domain labels. Unlike the above schemes, the proposed HyperDisGAN enforces the generated samples being close to the target samples in hyperplane space, which contributes to reshaping the hyperplane.

**Image Generation Combining Classifiers.** Starting with cGAN (Mirza & Osindero, 2014), auxiliary information such as class labels has been integrated with GANs to generate samples of a given specific type. SGAN (Odena, 2016) combines an auxiliary classifier with a discriminator to improve the generation performance. Several studies introduce an auxiliary classifier to reconstruct the auxiliary information from the generated samples, such as ACGAN (Odena et al., 2017) and VACGAN (Bazrafkan & Corcoran, 2018). Later, this idea is expanded to cross-domain image transformation by reconstructing target domain labels, such as conditional CycleGANs (Lu et al., 2018; Horita et al., 2018) and StarGAN (Choi et al., 2018). The auxiliary information like class labels reconstructed in these methods cannot reflect the variation degrees of generated samples, whereas the proposed HyperDisGAN explores the location of generated samples to control the variation degrees.

## 3 PROPOSED METHOD

### 3.1 FORMULATION

Our goal is to learn cross-domain mapping functions between $X$, and $Y$ and intra-domain mapping functions inside $X$ and $Y$ respectively, given training samples $\{x_i\}_{i=1}^N$ where $x_i \in X$, and $\{y_j\}_{j=1}^M$ where $y_j \in Y$. We denote the data distribution $x \sim P_X(x)$ and $y \sim P_Y(y)$, hyperplane vertical distance $d_x^v \in V_X$ and $d_y^v \in V_Y$ and hyperplane horizontal distance $d_x^h \in H_X$ and $d_y^h \in H_Y$. As illustrated in Figure 2, our model includes four mappings $G_{X2Y} : \{X, V_Y\} \to Y$, $G_{Y2X} : \{Y, V_X\} \to X$, $G_{X2X} : \{X, H_X\} \to X$ and $G_{Y2Y} : \{Y, H_Y\} \to Y$, where the generators generates images conditioned on both source image and target images' hyperplane distances. In addition, we introduce four adversarial discriminators $D_{X2Y}$, $D_{Y2X}$, $D_{X2X}$ and $D_{Y2Y}$, where $D_{X2Y}$ aims to discriminate between real images $\{y\}$ and generated images $\{G_{X2Y}(x, d_y^v)\}$; in the same way, $D_{Y2X}$ aims to discriminate between $\{x\}$ and $\{G_{Y2X}(y, d_x^v)\}$. $D_{X2X}$ aims to discriminate between real images $\{x\}$ and $\{G_{X2X}(x, d_x^h)\}$; in the same way, $D_{Y2Y}$ aims to discriminate between $\{y\}$ and $\{G_{Y2Y}(y, d_y^h)\}$.

### 3.2 HINGE LOSS AND HYPERPLANE DISTANCES

We pre-train an auxiliary classifier by *Hinge Loss* (Rosasco et al., 2004) to obtain an optimal hyperplane dividing the two classes samples. In the hyperplane space, the generated cross-domain samples not crossing over the opposite domain's minimized margin boundary need to be penalized, and the generated intra-domain samples crossing over the inside domain's minimized margin boundary also need to be penalized. Given a linear binary classifier $C(z) = w^T z + b$ which is trained on a training set $\{z_i, c_i\}_{i=1}^N$, $z_i \in R^D$, $c_i \in \{-1, +1\}$, the hinge loss can be expressed as follows:

$$\mathcal{L}_{\text{hinge}}(c_i, C(z_i)) = \frac{1}{N} \sum_{n=1}^N max[0, 1 - c_i(w^T z_i + b))]. \tag{1}$$

The vertical distances are considering as the controllable parameters to develop the generators $G_{X2Y}$ and $G_{Y2X}$ enabling cross-domain transformation between domain $X$ and domain $Y$. To measure the vertical distances from target samples to the optimal hyperplane, we introduce an auxiliary classifier $C_{aux}(z) = w_{aux}^T z + b_{aux}$ which is pre-trained via Equation 1. Specifically, given a random sample $x \in X$ and a random target sample $y \in Y$, the vertical distances $|d_v(x)|$ and $|d_v(y)|$ from the samples to the optimal hyperplane ($w_{aux}^T z + b_{aux} = 0$) are defined as follows, respectively:

$$|d_v(x)| = |C_{aux}(x)| = |w_{aux}^T x + b_{aux}| \in V_X, \tag{2}$$

$$|d_v(y)| = |C_{aux}(y)| = |w_{aux}^T y + b_{aux}| \in V_Y. \tag{3}$$

The horizontal distances are considering as the controllable parameters to develop the generators $G_{X2X}$ and $G_{Y2Y}$ enabling intra-domain transformation in domain $X$ and domain $Y$, respectively.

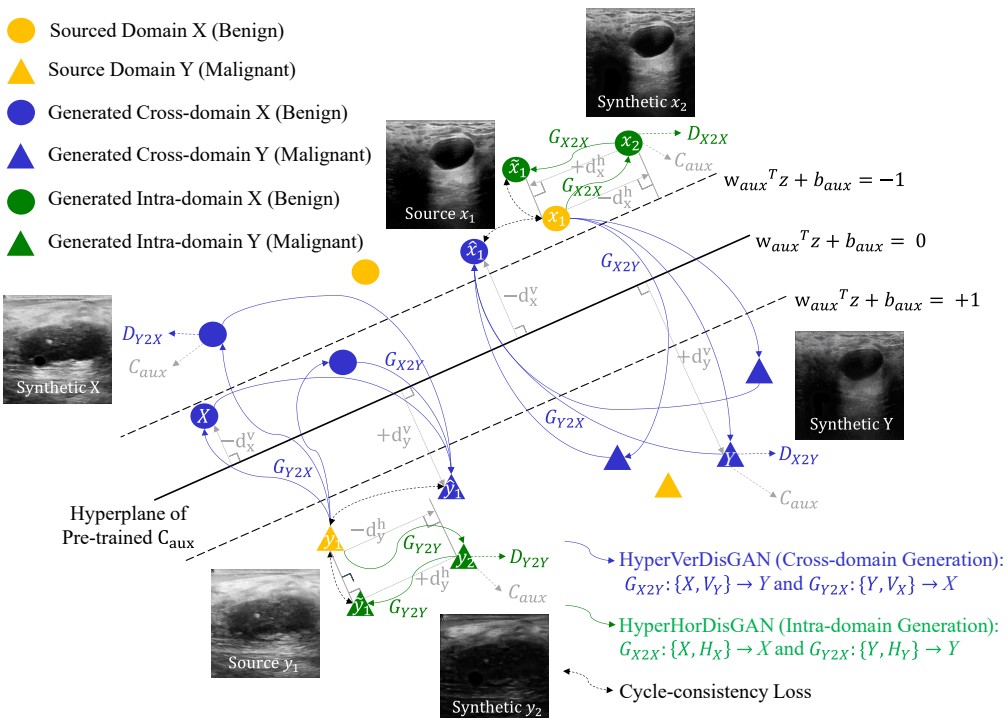

Figure 2: Overview of HyperDisGAN: HyperVerDisGAN controls the vertical distance for generated samples whereas HyperHorDisGAN controls the horizontal distance for generated samples. The target samples' distances are taken as the controllable parameters in the forward path. The $C_{aux}$ reconstructs the distances from the generated samples. The source samples' distances are are taken as controllable parameters to reconstruct the source samples in the backward path.

To measure the horizontal distances from the source samples to target samples in each intra-domain, we firstly obtain the coordinate distances from the source samples to target samples. The coordinates are represented by the vectors extracted before the last fully connection layer of the pre-trained auxiliary classifier $C_{aux}$. Specifically, given a random source sample $x_1 \in X$ and a random target sample $x_2 \in X$ with extracted vector $[coor_1^{x_1}, coor_2^{x_1}, \cdots, coor_m^{x_1}]$ and vector $[coor_1^{x_2}, coor_2^{x_2}, \cdots, coor_m^{x_2}]$ respectively, the coordinate distance $d_{coor}(x_1, x_2)$ between them is as follows:

$$d_{coor}(x_1, x_2) = d_{vector}(C_{aux}(x_1), C_{aux}(x_2)) = \sqrt{\sum_{i=1}^{m}(coor_i^{x_1} - coor_i^{x_2})^2}, \quad (4)$$

Similarly, the coordinate distance between random source $y_1$ and random target $y_2$ is as follows:

$$d_{coor}(y_1, y_2) = d_{vector}(C_{aux}(y_1), C_{aux}(y_2)) = \sqrt{\sum_{i=1}^{m}(coor_i^{y_1} - coor_i^{y_2})^2}, \quad (5)$$

We secondly obtain the differences of their vertical distances for the source samples and the target samples. Finally, the horizontal distances of the source samples and the target samples are calculated by *Pythagorean Theorem*. Given random samples $x_1, x_2 \in X$, and random samples $y_1, y_2 \in Y$, their horizontal distances $d_h(x_1, x_2)$ and $d_h(y_1, y_2)$ are as follows, respectively:

$$d_h(x_1, x_2) = \sqrt{d_{coor}(x_1, x_2)^2 - (|d_v(x_1)| - |d_v(x_2)|)^2} \in H_X, \quad (6)$$

$$d_h(y_1, y_2) = \sqrt{d_{coor}(y_1, y_2)^2 - (|d_v(y_1)| - |d_v(y_2)|)^2} \in H_Y. \quad (7)$$

### 3.3 HYPERDISGAN AND DOWNSTREAM CLASSIFICATION

Figure 2 illustrates the overall architecture of HyperDisGAN. Firstly, $G_{X2Y}$ transforms a source sample $x_1$ into $G_{X2Y}(x_1, +d_v(y))$, and $D_{X2Y}$ distinguishes between this transformed sample

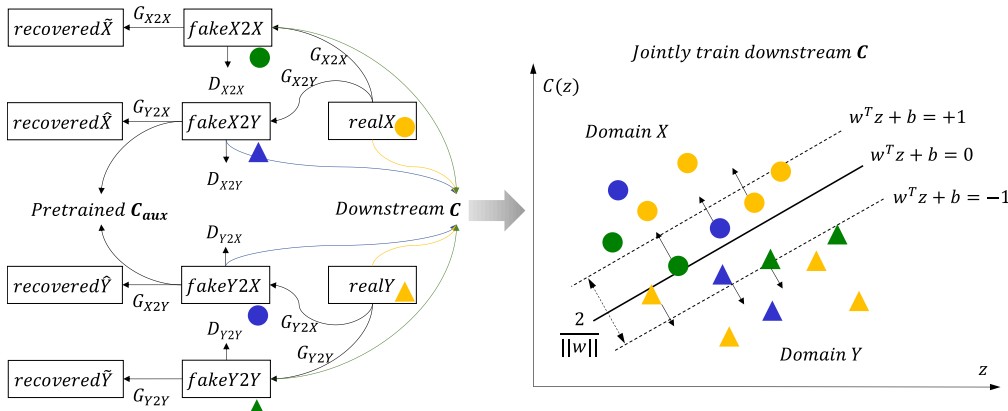

Figure 3: Overview of the data augmentation manner. The cross-domain and intra-domain samples which are generated by HyperVerDisGAN and HyperHorDisGAN respectively, are combined with the real samples to jointly train the downstream binary classifier via hinge loss.

and a real sample $y$, and vice versa for $y_1$, $G_{Y2X}$ and $D_{Y2X}$. $G_{X2X}$ transforms the source sample $x_1$ into $G_{X2X}(x_1, -d_h(x_1, x_2))$, and $D_{X2X}$ distinguishes between this transformed sample and a real sample $x_2$, and vice versa for $y_1$, $G_{Y2Y}$ and $D_{Y2Y}$. Secondly, a pre-trained auxiliary classifier $C_{aux}$ reconstructs the vertical distances $+d_v(y)$ and $-d_v(x)$ from the cross-domain generated samples $G_{X2Y}(x_1, +d_v(y))$ and $G_{Y2X}(y_1, -d_v(x))$, respectively; and reconstructs the horizontal distances $-d_h(x_1, x_2)$ and $-d_h(y_1, y_2)$ from the intra-domain generated samples $G_{X2X}(x_1, -d_h(x_1, x_2))$ and $G_{Y2Y}(y_1, -d_h(y_1, y_2))$, respectively. Thirdly, $G_{Y2X}$ inversely transforms the $G_{X2Y}(x_1, +d_v(y))$ into the reconstructed sample $\hat{x}_1$ using the vertical distance $-d_v(x_1)$, and vice versa for obtaining the reconstructed $\hat{y}_1$. $G_{X2X}$ inversely transforms the $G_{X2X}(x_1, -d_h(x_1, x_2))$ into the reconstructed sample $\tilde{x}_1$ using the horizontal distance $+d_h(x_1, x_2)$, and vice versa for obtaining the reconstructed $\tilde{y}_1$. We describe the detailed architectures used in our experiments in Appendix D.

The proposed HyperDisGAN contributes to the downstream classifier as an augmentation tool. As shown in Figure 3, the downstream classifier $C$ receives the real samples (*i.e., realX* and *realY*), cross-domain generated samples (*i.e., fakeX2Y* and *fakeY2X*) and intra-domain generated samples (*i.e., fakeX2X* and *fakeY2Y*). The pre-trained classifier $C_{aux}$ makes HyperDisGAN capable of controlling the locations of generated samples in vertical and horizontal directions. The cross-domain generated samples are thought to easily crossing over the optimal hyperplane, whereas the intra-domain generated samples are thought to fill their respective domain spaces. Therefore, the class labels of generated samples are defined to be the same with that of the target samples. The downstream classifier can be then optimized using hinge loss on the real and generated samples. Appendix H show the classification loss curves for CycleGAN and HyperDisGAN, respectively.

### 3.4 Objective Functions Of HyperDisGAN

Cross-domain and intra-domain generation are individual using two different objective functions.

To make the generated samples be indistinguishable from the real samples, we apply LSGAN loss (Mao et al., 2017) to cross-domain bidirectional mappings including $G_{X2Y} : X \rightarrow Y$ and $G_{Y2X} : Y \rightarrow X$, and intra-domain bidirectional mappings including $G_{X2X} : X \rightarrow X$ and $G_{Y2Y} : Y \rightarrow Y$. The LSGAN losses for $G_{X2Y}$ and $G_{X2X}$ are expressed as follows, respectively:

$$\mathcal{L}_{\text{crossGAN}}(G_{X2Y}, D_{X2Y}, X, Y) = \mathbb{E}_{y \sim P_Y}[(D_{X2Y}(y) - 1)^2] \\ + \mathbb{E}_{x_1 \sim P_X}[D_{X2Y}(G_{X2Y}(x_1, +d_v(y)))^2], \tag{8}$$

$$\mathcal{L}_{\text{intraGAN}}(G_{X2X}, D_{X2X}, X) = \mathbb{E}_{x_2 \sim P_X}[(D_{X2X}(x_2) - 1)^2] \\ + \mathbb{E}_{x_1 \sim P_X}[D_{X2X}(G_{X2X}(x_1, -d_h(x_1, x_2)))^2]. \tag{9}$$

and vice versa for the cross-domain mapping $G_{Y2X}$ and the intra-domain mapping $G_{Y2Y}$.

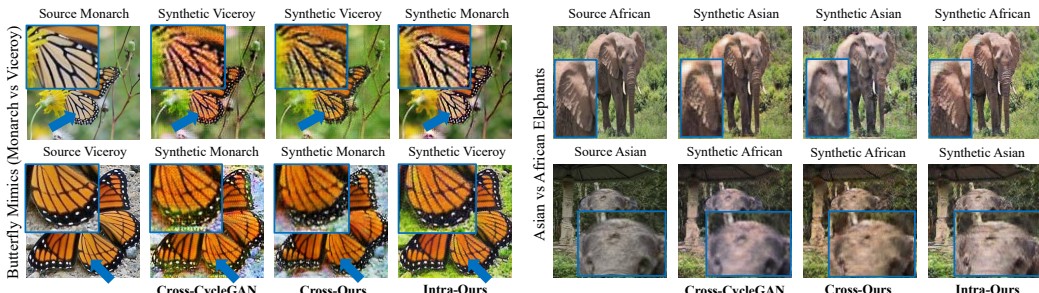

Figure 4: Qualitative results over Butterfly Mimics (KeithPinson, 2022) and Asian vs African Elephants (Goumiri et al., 2023). The generated images by the HyperDisGAN are clearly more realistic than that by the CycleGAN (Zhu et al., 2017). The location parameters for generating these images are provided by the auxiliary classifier ConvNeXt (Liu et al., 2022).

To control the generated samples' locations to be close as the target samples, we design the vertical distance loss and horizontal distance loss for cross-domain generation and intra-domain generation respectively. We calculate the vertical distances and horizontal distances for the generated samples according to Equations 2, 3 and 6, 7. The two types distance losses are formulated as follows:

$$
\begin{aligned}
\mathcal{L}_{\text{verDIS}}(G_{X2Y}, G_{Y2X}, C_{aux}) = & \ \mathbb{E}_{x_1 \sim P_X}[\||d_v(y)| - |d_v(G_{X2Y}(x_1, +d_v(y)))|\|_2^2] \\
& + \mathbb{E}_{y_1 \sim P_Y}[\||d_v(x)| - |d_v(G_{Y2X}(y_1, -d_v(x)))|\|_2^2],
\end{aligned}
\tag{10}
$$

$$
\begin{aligned}
\mathcal{L}_{\text{horDIS}}(&G_{X2X}, G_{Y2Y}, C_{aux}) \\
& = \mathbb{E}_{x_1 \sim P_X}[\|d_h(x_1, x_2) - d_h(x_1, G_{X2X}(x_1, -d_h(x_1, x_2)))\|_2^2] \\
& + \mathbb{E}_{y_1 \sim P_Y}[\|d_h(y_1, y_2) - d_h(y_1, G_{Y2Y}(y_1, -d_h(y_1, y_2)))\|_2^2].
\end{aligned}
\tag{11}
$$

For enforcing samples not to lose the original information after transforming twice, we develop the cycle-consistency loss $\mathcal{L}_{\text{crossCYC}}$ for cross-domain generation, and the cycle-consistency loss $\mathcal{L}_{\text{intraCYC}}$ for intra-domain generation:

$$
\begin{aligned}
\mathcal{L}_{\text{crossCYC}}(G_{X2Y}, G_{Y2X}) = & \ \mathbb{E}_{x_1 \sim P_X}[\|x_1 - G_{Y2X}(G_{X2Y}(x_1, +d_v(y)), -d_v(x_1))\|_1] \\
& + \mathbb{E}_{y_1 \sim P_Y}[\|y_1 - G_{X2Y}(G_{Y2X}(y_1, -d_v(x)), +d_v(y_1))\|_1],
\end{aligned}
\tag{12}
$$

$$
\begin{aligned}
\mathcal{L}_{\text{intraCYC}}(&G_{X2X}, G_{Y2Y}) \\
& = \mathbb{E}_{x_1 \sim P_X}[\|x_1 - G_{X2X}(G_{X2X}(x_1, -d_h(x_1, x_2)), +d_h(x_1, x_2))\|_1] \\
& + \mathbb{E}_{y_1 \sim P_Y}[\|y_1 - G_{Y2Y}(G_{Y2Y}(y_1, -d_h(y_1, y_2)), +d_h(y_1, y_2))\|_1].
\end{aligned}
\tag{13}
$$

Finally, the objective functions of HyperDisGAN for cross-domain generation and intra-domain generation are formulated as follows, respectively:

$$
\begin{aligned}
\mathcal{L}_{\text{HyperVerDisGAN}}(&G_{X2Y}, G_{Y2X}, C_{aux}, D_{X2Y}, D_{Y2X}) \\
& = \mathcal{L}_{\text{crossGAN}}(G_{X2Y}, D_{X2Y}, X, Y) + \mathcal{L}_{\text{crossGAN}}(G_{Y2X}, D_{Y2X}, Y, X) \\
& + \lambda_{\text{verDIS}}\mathcal{L}_{\text{verDIS}}(G_{X2Y}, G_{Y2X}, C_{aux}) \\
& + \lambda_{\text{crossCYC}}\mathcal{L}_{\text{crossCYC}}(G_{X2Y}, G_{Y2X}),
\end{aligned}
\tag{14}
$$

$$
\begin{aligned}
\mathcal{L}_{\text{HyperHorDisGAN}}(&G_{X2X}, G_{Y2Y}, C_{aux}, D_{X2X}, D_{Y2Y}) \\
& = \mathcal{L}_{\text{intraGAN}}(G_{X2X}, D_{X2X}, X, Y) + \mathcal{L}_{\text{intraGAN}}(G_{Y2Y}, D_{Y2Y}, Y, X) \\
& + \lambda_{\text{horDIS}}\mathcal{L}_{\text{horDIS}}(G_{X2Y}, G_{Y2X}, C_{aux}) \\
& + \lambda_{\text{intraCYC}}\mathcal{L}_{\text{intraCYC}}(G_{X2X}, G_{Y2Y}).
\end{aligned}
\tag{15}
$$

where $\lambda_{\text{verDIS}}, \lambda_{\text{crossCYC}}, \lambda_{\text{horDIS}}, \lambda_{\text{intraCYC}} > 0$ are some hyper-parameters balancing the losses. See Appendix B for detailed hyper-parameters settings for our experiments.

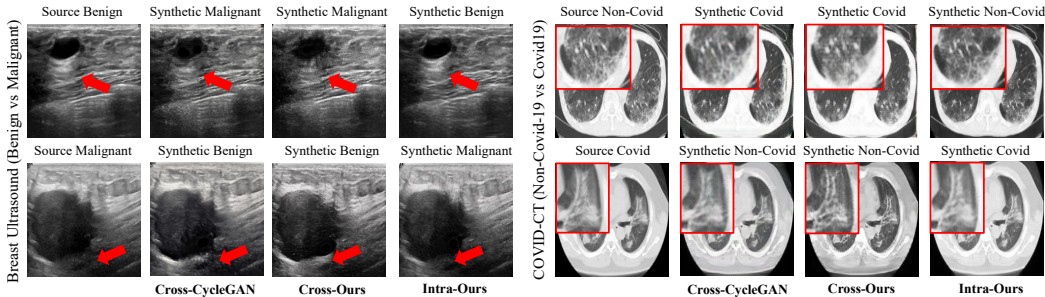

Figure 5: Qualitative results over mixed Breast Ultrasound (Al-Dhabyani et al., 2020; Yap et al., 2018) and COVID-CT (Zhao et al., 2020). The generated images by the HyperDisGAN is clearly more realistic than that by the CycleGAN (Zhu et al., 2017). The location parameters for generating these images are provided by the auxiliary classifier ConvNeXt (Liu et al., 2022).

| Datasets | Methods | Metrics | AlexNet | VGG13 | VGG16 | GoogleNet | ResNet18 | ResNet34 | DenseNet121 | MnasNet_0 | MobileNet_V3 | EfficientNet_V1 | ConvNeXt |
|---|---|---|---|---|---|---|---|---|---|---|---|---|---|
| Butterfly Mimics (Monarch vs Viceroy) | Original | ACC | 0.817±0.009 | 0.913±0.014 | 0.899±0.014 | 0.889±0.055 | 0.857±0.048 | 0.802±0.050 | 0.857±0.000 | 0.921±0.014 | 0.833±0.041 | 0.928±0.024 | 0.928±0.024 |
| | | AUC | 0.921±0.023 | 0.959±0.004 | 0.955±0.007 | 0.945±0.011 | 0.941±0.018 | 0.923±0.020 | 0.927±0.034 | 0.940±0.005 | 0.927±0.025 | 0.958±0.015 | 0.943±0.012 |
| | Traditional Augment (TA) | ACC | 0.857±0.024 | 0.905±0.000 | 0.929±0.000 | 0.881±0.000 | 0.857±0.041 | 0.841±0.036 | 0.897±0.014 | 0.897±0.027 | 0.865±0.014 | 0.937±0.014 | 0.929±0.000 |
| | | AUC | 0.910±0.018 | 0.946±0.008 | 0.940±0.014 | 0.945±0.003 | 0.931±0.025 | 0.931±0.022 | 0.954±0.015 | 0.938±0.026 | 0.926±0.015 | 0.960±0.006 | 0.960±0.008 |
| | TA+ACGAN | ACC | 0.857±0.041 | 0.921±0.014 | 0.921±0.014 | 0.881±0.000 | 0.865±0.028 | 0.770±0.027 | 0.897±0.013 | 0.921±0.013 | 0.778±0.027 | 0.913±0.027 | 0.929±0.000 |
| | | AUC | 0.924±0.018 | 0.964±0.003 | 0.959±0.015 | 0.941±0.017 | 0.942±0.026 | 0.901±0.043 | 0.966±0.009 | 0.936±0.004 | 0.891±0.034 | 0.955±0.016 | 0.953±0.012 |
| | TA+VACGAN | ACC | 0.802±0.050 | 0.905±0.000 | 0.936±0.014 | 0.857±0.041 | 0.825±0.055 | 0.722±0.060 | 0.857±0.041 | 0.897±0.014 | 0.802±0.059 | 0.873±0.077 | 0.913±0.014 |
| | | AUC | 0.876±0.051 | 0.958±0.005 | 0.950±0.008 | 0.933±0.001 | 0.897±0.045 | 0.864±0.048 | 0.936±0.025 | 0.936±0.006 | 0.883±0.043 | 0.930±0.018 | 0.959±0.004 |
| | TA+CycleGAN | ACC | 0.802±0.060 | 0.929±0.000 | 0.913±0.014 | 0.897±0.014 | 0.833±0.063 | 0.794±0.027 | 0.873±0.050 | 0.881±0.024 | 0.841±0.027 | 0.905±0.041 | 0.913±0.014 |
| | | AUC | 0.924±0.004 | 0.942±0.009 | 0.962±0.014 | 0.948±0.006 | 0.911±0.034 | 0.915±0.001 | 0.919±0.024 | 0.938±0.009 | 0.913±0.017 | 0.944±0.035 | 0.936±0.005 |
| | TA+HyperDisGAN | ACC | 0.881±0.024 | 0.921±0.014 | 0.929±0.024 | 0.841±0.027 | 0.865±0.036 | 0.873±0.028 | 0.905±0.024 | 0.926±0.014 | 0.857±0.014 | 0.929±0.000 | 0.937±0.014 |
| | | AUC | 0.932±0.014 | 0.964±0.003 | 0.963±0.007 | 0.952±0.014 | 0.945±0.006 | 0.938±0.015 | 0.969±0.009 | 0.938±0.010 | 0.927±0.005 | 0.961±0.006 | 0.966±0.006 |
| Asian vs African Elephants | Original | ACC | 0.750±0.017 | 0.842±0.025 | 0.832±0.059 | 0.797±0.051 | 0.802±0.003 | 0.727±0.014 | 0.748±0.023 | 0.863±0.006 | 0.848±0.038 | 0.885±0.013 | 0.900±0.017 |
| | | AUC | 0.845±0.028 | 0.916±0.010 | 0.897±0.050 | 0.873±0.032 | 0.859±0.014 | 0.806±0.040 | 0.850±0.036 | 0.933±0.007 | 0.920±0.009 | 0.921±0.005 | 0.948±0.031 |
| | Traditional Augment (TA) | ACC | 0.763±0.033 | 0.822±0.020 | 0.848±0.024 | 0.822±0.018 | 0.733±0.060 | 0.735±0.013 | 0.787±0.043 | 0.858±0.025 | 0.850±0.015 | 0.853±0.023 | 0.908±0.016 |
| | | AUC | 0.861±0.010 | 0.898±0.027 | 0.915±0.016 | 0.900±0.047 | 0.814±0.042 | 0.811±0.019 | 0.905±0.026 | 0.937±0.005 | 0.919±0.011 | 0.906±0.002 | 0.948±0.020 |
| | TA+ACGAN | ACC | 0.788±0.072 | 0.803±0.026 | 0.742±0.059 | 0.838±0.038 | 0.733±0.015 | 0.660±0.022 | 0.792±0.055 | 0.838±0.038 | 0.805±0.015 | 0.868±0.019 | 0.883±0.003 |
| | | AUC | 0.860±0.061 | 0.897±0.028 | 0.828±0.078 | 0.922±0.027 | 0.828±0.007 | 0.706±0.052 | 0.862±0.046 | 0.933±0.009 | 0.873±0.019 | 0.934±0.009 | 0.938±0.000 |
| | TA+VACGAN | ACC | 0.755±0.051 | 0.697±0.019 | 0.750±0.053 | 0.640±0.009 | 0.662±0.003 | 0.657±0.020 | 0.655±0.013 | 0.825±0.033 | 0.565±0.018 | 0.807±0.046 | 0.870±0.017 |
| | | AUC | 0.855±0.040 | 0.788±0.020 | 0.842±0.015 | 0.712±0.015 | 0.726±0.022 | 0.733±0.015 | 0.751±0.041 | 0.931±0.010 | 0.595±0.012 | 0.886±0.013 | 0.928±0.009 |
| | TA+CycleGAN | ACC | 0.822±.031 | 0.862±0.018 | 0.828±0.034 | 0.852±0.021 | 0.708±0.033 | 0.695±0.058 | 0.735±0.022 | 0.863±0.006 | 0.833±0.003 | 0.767±0.012 | 0.918±0.006 |
| | | AUC | 0.896±0.010 | 0.918±0.019 | 0.896±0.014 | 0.924±0.013 | 0.775±0.018 | 0.743±0.060 | 0.790±0.022 | 0.940±0.003 | 0.919±0.008 | 0.843±0.013 | 0.966±0.006 |
| | TA+HyperDisGAN | ACC | 0.830±0.019 | 0.838±0.014 | 0.867±0.009 | 0.865±0.008 | 0.815±0.032 | 0.748±0.032 | 0.832±0.005 | 0.848±0.014 | 0.837±0.021 | 0.886±0.032 | 0.925±0.006 |
| | | AUC | 0.898±0.005 | 0.920±0.001 | 0.925±0.008 | 0.924±0.001 | 0.886±0.010 | 0.838±0.028 | 0.917±0.018 | 0.944±0.007 | 0.921±0.005 | 0.927±0.015 | 0.969±0.004 |

Table 1: Comparisons with the state of the arts over Butterfly Mimics and Asian vs African Elephants: Training with 90 (Monarch Butterfly), 67 (Viceroy Butterfly), 294 (Asian Elephant), and 296 (African Elephant) samples, the proposed augmentation manner (TA+HyperDisGAN) mostly performs the best. We report ACC (↑) and AUC (↑) averaged over three runs.

# 4 EXPERIMENTS AND RESULTS

## 4.1 EXPERIMENTS ON BUTTERFLY MIMICS AND ELEPHANTS

Figure 4 qualitatively demonstrates that the HyperDisGAN outperforms the cross-domain transformation technique (*i.e.,* CycleGAN) in data-limited generation, especially in terms of the generated textures and shapes. The HyperDisGAN is prone to generate the black postmedian stripe across hindwing for viceroy butterflies and eliminate it in the opposite generation direction. The HyperDisGAN generates asian elephants with light gray and african elephants with grayish brown. Moreover, it clears two bumps in the top of the source asian elephant's head. See Appendix E for the datasets description for our experiments.

Table 1 compares the proposed augmentation manner with the state-of-the-art methods in data-limited natural image classification over the Butterfly Mimics and Asian vs African Elephants. We can see that our augmentation manner mostly performs the best in AUC score, demonstrating its effectiveness of improving the downstream classifiers' generalization. Specially, transfer learning based on ImageNet Database (Deng et al., 2009) is used to pre-train the auxiliary classifiers in the VACGAN and the HyperDisGAN. Each row (TA) also reflects the pre-trained auxiliary classifiers. Ablation study of HyperDisGAN's two major components is in Appendix C.

| Datasets | Methods | Metrics | AlexNet | VGG13 | VGG16 | GoogleNet | ResNet18 | ResNet34 | DenseNet121 | MnasNet1_0 | MobileNet_V3 | EfficientNet_V1 | ConvNeXt |
|---|---|---|---|---|---|---|---|---|---|---|---|---|---|
| Breast Ultrasound (Benign vs Malignant) | Original | ACC | 0.833±0.005 | 0.824±0.033 | 0.821±0.000 | 0.830±0.028 | 0.808±0.038 | **0.830±0.009** | 0.789±0.024 | 0.852±0.005 | 0.774±0.053 | 0.843±0.014 | 0.852±0.029 |
| | | AUC | **0.931±0.011** | 0.916±0.023 | 0.910±0.014 | 0.910±0.013 | 0.898±0.026 | 0.910±0.013 | 0.892±0.024 | 0.925±0.007 | 0.843±0.042 | 0.933±0.007 | 0.906±0.020 |
| | Traditional Augment (TA) | ACC | 0.805±0.027 | 0.833±0.005 | 0.763±0.025 | 0.811±0.052 | 0.802±0.025 | 0.815±0.038 | 0.802±0.028 | 0.824±0.045 | 0.780±0.063 | 0.865±0.011 | 0.862±0.005 |
| | | AUC | 0.897±0.008 | 0.925±0.013 | 0.907±0.002 | 0.887±0.048 | 0.912±0.019 | 0.900±0.015 | 0.891±0.028 | 0.920±0.011 | 0.851±0.060 | 0.915±0.012 | 0.935±0.005 |
| | TA+ACGAN | ACC | 0.821±0.000 | 0.818±0.033 | 0.755±0.050 | 0.774±0.059 | 0.802±0.057 | 0.783±0.009 | 0.792±0.016 | 0.840±0.019 | 0.770±0.024 | 0.774±0.087 | 0.874±0.011 |
| | | AUC | 0.917±0.025 | 0.908±0.042 | 0.867±0.020 | 0.875±0.027 | 0.866±0.032 | 0.867±0.036 | 0.896±0.003 | 0.913±0.007 | 0.899±0.055 | 0.853±0.089 | 0.949±0.003 |
| | TA+VACGAN | ACC | 0.805±0.005 | 0.814±0.029 | 0.774±0.049 | 0.758±0.044 | 0.638±0.005 | 0.767±0.093 | 0.641±0.086 | 0.830±0.028 | 0.808±0.020 | 0.704±0.080 | 0.868±0.016 |
| | | AUC | 0.920±0.003 | 0.899±0.008 | 0.885±0.017 | 0.854±0.014 | 0.749±0.021 | 0.847±0.046 | 0.813±0.063 | 0.932±0.014 | 0.883±0.018 | 0.840±0.022 | 0.939±0.010 |
| | TA+CycleGAN | ACC | 0.805±0.005 | **0.865±0.014** | 0.821±0.025 | 0.811±0.028 | **0.814±0.005** | 0.805±0.014 | 0.774±0.041 | 0.833±0.014 | 0.805±0.022 | 0.849±0.016 | 0.874±0.005 |
| | | AUC | 0.894±0.015 | 0.929±0.004 | 0.922±0.018 | 0.914±0.009 | 0.894±0.011 | 0.889±0.019 | 0.867±0.013 | 0.911±0.017 | 0.901±0.021 | 0.919±0.011 | 0.946±0.006 |
| | TA+HyperDisGAN | ACC | **0.859±0.019** | 0.862±0.005 | **0.830±0.011** | **0.852±0.014** | 0.811±0.000 | 0.821±0.028 | **0.837±0.014** | **0.886±0.014** | **0.876±0.061** | **0.878±0.014** | **0.887±0.009** |
| | | AUC | **0.931±0.001** | **0.946±0.007** | **0.929±0.010** | **0.919±0.005** | **0.916±0.008** | **0.920±0.003** | **0.922±0.016** | **0.934±0.007** | **0.933±0.013** | **0.940±0.010** | **0.953±0.009** |
| COVID-CT (Non-Covid-19 vs Covid-19) | Original | ACC | 0.734±0.020 | 0.770±0.024 | 0.765±0.014 | 0.691±0.058 | 0.688±0.033 | 0.713±0.021 | 0.727±0.012 | 0.777±0.027 | 0.777±0.022 | **0.783±0.039** | 0.777±0.030 |
| | | AUC | 0.793±0.013 | 0.856±0.002 | 0.842±0.015 | 0.764±0.017 | 0.772±0.024 | 0.770±0.003 | 0.818±0.018 | 0.830±0.020 | 0.844±0.039 | 0.831±0.042 | 0.868±0.020 |
| | Traditional Augment (TA) | ACC | 0.724±0.037 | **0.793±0.013** | 0.736±0.032 | 0.755±0.040 | 0.714±0.037 | 0.695±0.013 | 0.757±0.010 | **0.803±0.031** | 0.762±0.010 | 0.754±0.063 | 0.793±0.010 |
| | | AUC | 0.789±0.017 | 0.859±0.006 | 0.833±0.024 | 0.818±0.037 | 0.796±0.041 | 0.750±0.016 | 0.816±0.012 | 0.835±0.008 | 0.820±0.015 | 0.837±0.036 | 0.876±0.004 |
| | TA+ACGAN | ACC | 0.657±0.059 | 0.744±0.034 | 0.773±0.023 | 0.667±0.047 | 0.726±0.014 | 0.714±0.015 | 0.718±0.057 | 0.731±0.006 | 0.704±0.005 | 0.701±0.078 | 0.803±0.000 |
| | | AUC | 0.724±0.051 | 0.821±0.017 | 0.849±0.011 | 0.780±0.032 | 0.798±0.023 | 0.764±0.019 | 0.776±0.063 | 0.787±0.001 | 0.796±0.018 | 0.772±0.083 | 0.877±0.001 |
| | TA+VACGAN | ACC | 0.673±0.017 | 0.744±0.043 | 0.698±0.069 | 0.696±0.038 | 0.637±0.024 | 0.619±0.088 | 0.652±0.029 | 0.750±0.003 | 0.723±0.037 | 0.654±0.010 | 0.780±0.017 |
| | | AUC | 0.775±0.025 | 0.813±0.039 | 0.813±0.074 | 0.780±0.034 | 0.714±0.024 | 0.708±0.053 | 0.741±0.044 | 0.818±0.006 | 0.803±0.030 | 0.738±0.036 | 0.849±0.003 |
| | TA+CycleGAN | ACC | 0.709±0.027 | 0.773±0.032 | 0.757±0.033 | 0.765±0.006 | 0.726±0.049 | 0.732±0.032 | 0.742±0.037 | 0.771±0.006 | 0.759±0.026 | 0.745±0.019 | 0.775±0.022 |
| | | AUC | 0.784±0.010 | 0.851±0.018 | 0.830±0.034 | **0.856±0.011** | 0.815±0.034 | 0.791±0.025 | 0.805±0.032 | 0.835±0.002 | 0.817±0.033 | 0.819±0.016 | 0.838±0.012 |
| | TA+HyperDisGAN | ACC | **0.765±0.010** | 0.782±0.015 | **0.789±0.008** | **0.780±0.019** | **0.736±0.021** | **0.770±0.019** | **0.789±0.009** | 0.793±0.013 | **0.787±0.008** | 0.783±0.008 | **0.816±0.012** |
| | | AUC | **0.813±0.010** | **0.865±0.004** | **0.852±0.006** | 0.842±0.019 | **0.823±0.014** | **0.838±0.013** | **0.845±0.010** | **0.837±0.008** | **0.853±0.011** | **0.844±0.007** | **0.885±0.013** |

Table 2: Comparisons with the state of the arts over mixed Breast Ultrasound and COVID-CT: Training with 440 (Benign Breast Lesion), 158 (Malignant Breast Lesion), 234 (Non-Covid-19), and 191 (Covid-19) samples. The proposed augmentation manner (TA+HyperDisGAN) mostly outperforms the state-of-the-arts in AUC score on the limited medical image datasets. We report ACC (↑) and AUC (↑) averaged over three runs.

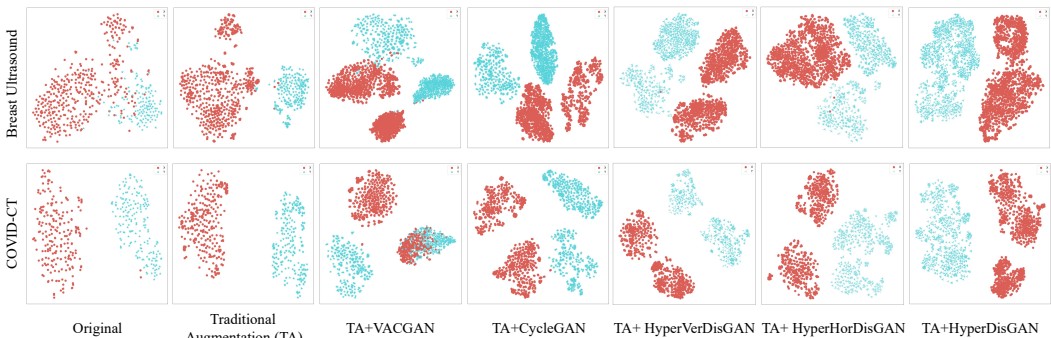

Figure 6: Training samples' distribution of downstream ConvNeXt (Liu et al., 2022) visualized by t-SNE. From left to right column: original samples, traditional augmentation (TA), TA+VACGAN, TA+CycleGAN, TA+HyperVerDisGAN, TA+HyperHorDisGAN, TA+HyperDisGAN.

## 4.2 EXPERIMENTS ON BREAST ULTRASOUND AND COVID-CT

Figure 5 demonstrates that the HyperDisGAN can generate more realistic images compared with the CycleGAN. The HyperDisGAN generates dark shadowing below the lesion when transform the source benign images, and generate bright region in the inverse transformation. Moreover, we can observe the smoothness changes of the breast lesion's boundary. In bi-directional generation of CT images, we can also see the generation and elimination of ground glass shadow regions.

Table 2 quantitatively compares the proposed augmentation manner with several state-of-the-art GANs over datasets Mixed Breast Ultrasound and COVID-CT. We can see that this augmentation manner consistently outperforms traditional-only augmentation and mostly outperforms the state-of-the-arts in AUC score, especially using the limited training samples. Appendix C shows the ablation study for HyperDisGAN's two major components.

## 4.3 VISUALIZATION OF TRAINING SAMPLE DISTRIBUTION

The generated samples participate in reshaping the hyperplane in the latent space. Figure 6 shows the training samples' distribution and the reshaped hyperplanes of various methods on two limited medical datasets by t-Distributed Stochastic Neighbor Embedding (t-SNE) (Van der Maaten & Hin-

ton, 2008), by downstream ConvNeXt. Compared with the state of the arts, TA+HyperDisGAN can construct a more precise hyperplane by generating samples alongside the its margins. Training samples' distribution for more downstream classification models are shown in Appendix G.

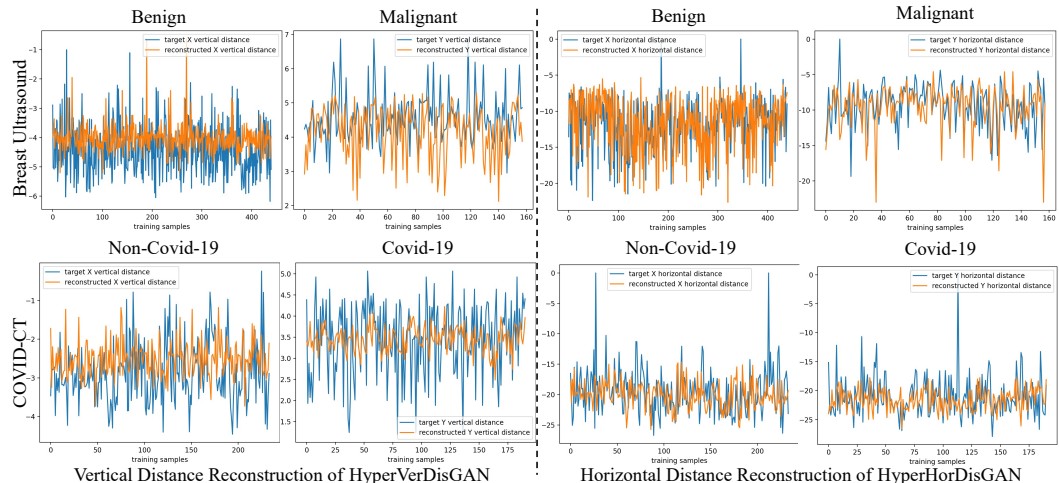

Figure 7: The curves of vertical and horizontal distance (Blue) and the auxiliary classifier's reconstructed distance (Orange). The examples comes from an auxiliary classifier ConvNeXt.

## 5 DISCUSSION

**Innovation of parameterized network:** To the best of our knowledge, we are the first to collect sample's location information in the initial classification phase and generating controllable samples reshaping the decision boundary in the second classification phase (same classification architecture as the initial phase). The collected location information includes the vertical distances between the samples and the optimal hyperplane, and the horizontal distances between the intra-domain samples.

The SGAN and ACGAN's auxiliary classifiers share the same architectures with the discriminators, whereas the HyperDisGAN's auxiliary classifier is external and can easily change to any latest architecture like VACGAN. Moreover, SGAN's auxiliary classifier takes the generated samples as an additional class, and ACGAN and VACGAN's auxiliary classifiers reconstruct the classification labels from the generated samples. Unlike these methods, the HyperDisGAN's auxiliary classifier reconstructs location information for the cross-domain and intra-domain samples.

**Effectiveness of parameterized network:** The success of the HyperDisGAN is the controllable variation degrees represented by the parameterized vertical and horizontal distances. The ability of the parameterized generators are shown in Figure 7 using two limited medical datasets: the reconstructed vertical distance curve trend is generally consistent with the target's vertical distance curve (left); and the reconstructed horizontal distance curve trend is consistent with the target's horizontal distance curve (right). More experimental results are presented in Appendix F. Therefore, controlling the location of generated samples can be achieved by the parameterized generation, whereas the previous GAN-based methods are difficult to achieve this goal when training data is extremely small like the tiny Butterfly Mimics dataset.

## 6 CONCLUSION

The primary aim of this paper is to develop a general data augmentation manner that can be applied to different classifiers' architectures. We have showcased the effectiveness of the HyperDisGAN for generating diverse samples to improve the eleven downstream classification models in both natural and medical datasets. We observe that enabling controlling the variation degrees of generated samples has significantly more impact than blindly increasing the number of the training samples.

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

## A    BASELINE OF CYCLEGAN

According to the basic formulation in CycleGAN Zhu et al. (2017), given two domains $X$ and $Y$, we consider unpaired training samples $\{x_i\}_{i=1}^N$ where $x_i \in X$, and $\{y_i\}_{i=1}^N$ where $y_i \in Y$. The goal of unpaired image-to-image transformation is to learn bidirectional mappings including $G_{X2Y} : X \to Y$ and $G_{Y2X} : Y \to X$. Adversarial discriminators $D_X$ and $D_Y$ are employed to distinguish between real images and generated images. In particular, the $D_X$ aims to discriminate between real images $\{x\}$ and generated images $\{G_{Y2X}(y)\}$; similarly, $D_Y$ discriminates between $\{y\}$ and $\{G_{X2Y}(x)\}$.

Therefore, the adversarial losses for both mappings are expressed as follows respectively:

$$\mathcal{L}_{\text{GAN}}(G_{X2Y}, D_Y, X, Y) = \mathbb{E}_{y \sim P_Y}[log D_Y(y)] + \mathbb{E}_{x \sim P_X}[log(1 - D_Y(G_{X2Y}(x))], \qquad (16)$$

$$\mathcal{L}_{\text{GAN}}(G_{Y2X}, D_X, Y, X) = \mathbb{E}_{x \sim P_X}[log D_X(x)] + \mathbb{E}_{y \sim P_Y}[log(1 - D_X(G_{Y2X}(y))]. \qquad (17)$$

Since the samples in the two domains are unpaired, cycle consistency is introduced to establish relationships between individual input $x_i$ and a desired output $y_i$. The cycle consistency enforces that $G_{X2Y}$ and $G_{Y2X}$ are a pair of inverse mappings, and that the transformed samples can be mapped back to the original samples. The cycle consistency includes the forward cycle consistency $x \to G_{X2Y}(x) \to G_{Y2X}(G_{X2Y}(x)) \approx x$ and the backward cycle consistency $y \to G_{Y2X}(y) \to G_{X2Y}(G_{Y2X}(y)) \approx y$. Thus, the cycle consistency loss is formulated as:

$$\begin{aligned}\mathcal{L}_{\text{cyc}}(G_{X2Y}, G_{Y2X}) = \mathbb{E}_{x \sim P_X}[||G_{Y2X}(G_{X2Y}(x) - x||_1] \\ + \mathbb{E}_{y \sim P_Y}[||G_{X2Y}(G_{Y2X}(y) - y||_1].\end{aligned} \qquad (18)$$

With the cycle consistency loss, the overall objective is written as:

$$\begin{aligned}\mathcal{L}(G_{X2Y}, G_{Y2X}, D_X, D_Y) = \mathcal{L}_{\text{GAN}}(G_{X2Y}, D_Y, X, Y) + \mathcal{L}_{\text{GAN}}(G_{Y2X}, D_X, Y, X) \\ + \lambda \mathcal{L}_{\text{cyc}}(G_{X2Y}, G_{Y2X}).\end{aligned} \qquad (19)$$

where the weight $\lambda$ determines the significance of the corresponding objective.

CycleGAN can be taken as an augmentation tool increasing the number of training samples by generating the arbitrary samples from one domain to another. Specifically, the generated samples are combined with the real samples to jointly train the downstream binary classifiers. However, the quality of generated samples with respect to their classification labels is uncertain, resulting in impairing the downstream classification. To address this issue, we control the variation degree for the generated samples in the both cross-domain and intra-domain generation background.

## B    TRAINING SETTINGS.

All networks are trained using Adam (Kingma & Ba, 2014) with $\beta_1 = 0.5$ and $\beta_2 = 0.999$. The initial learning rate is set to $1 \times e^{-4}$ over the first 25 epochs and linearly decays to 0 over the next 25 epochs. All images are normalized between -1 and 1, and resized to $224 \times 224$. For the auxiliary and downstream classifiers optimized by hinge loss, the number of output node of last linear layer is set to 1. We adopt transfer learning for all downstream classifiers using ImageNet Dataset (Deng et al., 2009). We use traditional augmentation such as horizontal flipping to pre-augment the training samples for the state of the arts and the HyperDisGAN.

We set the same downstream classification weights for the real samples and that of the generated samples. For fair comparison, we follow the CycleGAN (Zhu et al., 2017) for $\lambda_{\text{crossCYC}} = 10.0$, $\lambda_{\text{intraCYC}} = 10.0$. Considering the vertical distances and horizontal distances having different large-scales, the hyper-parameter $\lambda_{\text{verDIS}}$ is set from 0.01 to 0.1 and the hyper-parameter $\lambda_{\text{horDis}}$ is set from 0.001 to 0.01. The details of hyper-parameters are described in Table 3 for better reproducibility. Note that the $\lambda_{\text{horDIS}}$ is less than or equal to the $\lambda_{\text{verDIS}}$, and several $\lambda_{\text{horDIS}}$, $\lambda_{\text{verDIS}}$ of auxiliary classifiers are less or equal to that of other auxiliary classifiers. These settings have the following two reasons respectively: (1) The error of reconstructing the horizontal distances is larger than that of reconstructing the vertical distances. (2) Several specific auxiliary classifiers constructs overfitting hyperplanes resulting in the larger scales of the distances.

| Datasets | Hyper-parameters | AlexNet | VGG13 | VGG16 | GoogLeNet | ResNet18 | ResNet34 | DenseNet121 | MnasNet1_0 | MobileNet_V3 | EfficientNet_V1 | ConvNeXt |
|---|---|---|---|---|---|---|---|---|---|---|---|---|
| Butterfly Mimics | $\lambda_{\text{verDIS}}$ | 0.1 | 0.1 | 0.1 | 0.1 | 0.1 | 0.1 | 0.1 | 0.1 | 0.1 | 0.1 | 0.1 |
| | $\lambda_{\text{horDIS}}$ | 0.001 | 0.001 | 0.001 | 0.01 | 0.001 | 0.001 | 0.001 | 0.001 | 0.001 | 0.01 | 0.01 |
| Asian vs African Elephants | $\lambda_{\text{verDIS}}$ | 0.1 | 0.1 | 0.1 | 0.1 | 0.1 | 0.1 | 0.1 | 0.1 | 0.1 | 0.1 | 0.1 |
| | $\lambda_{\text{horDIS}}$ | 0.001 | 0.001 | 0.001 | 0.01 | 0.001 | 0.01 | 0.001 | 0.001 | 0.001 | 0.01 | 0.01 |
| Breast Ultrasound | $\lambda_{\text{verDIS}}$ | 0.1 | 0.01 | 0.01 | 0.1 | 0.1 | 0.1 | 0.01 | 0.1 | 0.01 | 0.01 | 0.1 |
| | $\lambda_{\text{horDIS}}$ | 0.001 | 0.001 | 0.001 | 0.001 | 0.001 | 0.001 | 0.001 | 0.001 | 0.001 | 0.001 | 0.01 |
| COVID-CT | $\lambda_{\text{verDIS}}$ | 0.1 | 0.01 | 0.01 | 0.1 | 0.1 | 0.1 | 0.01 | 0.1 | 0.01 | 0.1 | 0.1 |
| | $\lambda_{\text{horDIS}}$ | 0.001 | 0.001 | 0.001 | 0.001 | 0.001 | 0.001 | 0.001 | 0.001 | 0.001 | 0.01 | 0.01 |

Table 3: Hyper-parameters settings of the $\lambda_{\text{verDIS}}$ and the $\lambda_{\text{horDIS}}$ for the HyperDisGAN.

| DataSets | Metrics | Original | Traditional Augmentation (TA) | TA+VD | TA+HD | TA+HD+HD |
|---|---|---|---|---|---|---|
| Butterfly Mimics | ACC | 0.872 | 0.889 | 0.891 | 0.887 | 0.897 |
| | AUC | 0.934 | 0.937 | 0.946 | 0.940 | 0.951 |
| Asian vs African Elephants | ACC | 0.820 | 0.817 | 0.830 | 0.821 | 0.845 |
| | AUC | 0.892 | 0.892 | 0.896 | 0.908 | 0.915 |

(a) Butterfly Mimics and Asian vs African Elephants.

| DataSets | Metrics | Original | Traditional Augmentation (TA) | TA+VD | TA+HD | TA+HD+HD |
|---|---|---|---|---|---|---|
| Breast Ultrasound | ACC | 0.823 | 0.817 | 0.830 | 0.817 | 0.854 |
| | AUC | 0.908 | 0.905 | 0.920 | 0.907 | 0.931 |
| COVID-CT | ACC | 0.745 | 0.747 | 0.757 | 0.756 | 0.781 |
| | AUC | 0.820 | 0.822 | 0.831 | 0.829 | 0.845 |

(b) mixed Breast Ultrasound and COVID-CT.

Table 4: Ablation study of the HyperDisGAN: controllable vertical distance (VD) and controllable horizontal distance (HD). The HyperDisGAN performs the best as VD and HD are complementary to each other. The ACC (↑) and AUC (↑) averaged over the eleven downstream classifiers.

## C  ABLATION STUDY

The HyperDisGAN consists of two major components: the HyperVerDisGAN controlling the vertical distances (VDs) and the HyperHorDisGAN controlling the vertical distances (HDs). We evaluate the contributions of these two components to the overall downstream classification respectively. As illustrated in Table 4, the HyperDisGAN combining the VD and HD gets the highest ACC and AUC scores using the average performance of the eleven classification architectures.

## D  IMPLEMENTATION DETAILS

**Network Architecture.**  The generators adopt Johnson et al. (2016) containing three convolutions for downsampling, nine residual blocks, and two transposed convolutions with the stride size of $\frac{1}{2}$ for upsampling. We adopt instance normalization (Ulyanov et al., 2016) in the generators but no normalization in the first convolution of discriminator. We add one channel for the first convolutional layer, because the distance parameters needs to be spatially replicated match the size of the input image and concatenated with the input image. The discriminators adopt PatchGANs (Isola et al., 2017) to determine if the $70 \times 70$ overlapping image patches is a real one of a generated one.

**Classification Model Selection.**  We randomly select eleven downstream classifiers with different architectures including AlexNet (Krizhevsky et al., 2012), VGGs (Simonyan & Zisserman, 2015), GoogLeNet (Szegedy et al., 2015), ResNets (He et al., 2016), DenseNet (Huang et al., 2017), MnasNet 1_0 (Tan et al., 2019), MobileNet V3 (Small) (Howard et al., 2019), EfficientNet V1 (B5) (Tan & Le, 2019) and ConvNeXt (Tiny) (Liu et al., 2022). In the HyperDisGAN and the VACGAN, the auxiliary classifiers share the same deep learning architectures with the downstream classifiers.

## E  DATASETS AND EVALUATION METRICS

### E.1  DATASETS COLLECTION AND EVALUATION METRICS

We conduct experiments over multiple public datasets: Butterfly Mimics (KeithPinson, 2022), Asian vs African Elephants (Goumiri et al., 2023; Zhang et al., 2023), Breast Ultrasound Images Dataset (BUSI) (Al-Dhabyani et al., 2020), Breast Ultrasound images Collected from UDIAT Diagnostic Center (UDIAT) (Yap et al., 2018) and CT Scan Dataset about COVID-19 (COVID-CT) (Zhao et al., 2020), and perform evaluations with two widely adopted metrics in image classification: accuracy (ACC) and the area under the receiver operating characteristic curve (AUC). We mix the BUSI dataset with UDIAT dataset to increase the challenge of breast ultrasound image classification.

| Mode | Butterfly Mimics | | Asian vs African Elephants | | Breast Ultrasound | | COVID-CT | |
|---|---|---|---|---|---|---|---|---|
| | Monarch Butterfly | Viceroy Butterfly | Asian Elephant | African Elephant | Benign Lesion | Malignant Lesion | non-COVID-19 | COVID-19 |
| Train | 90 | 67 | 294 | 296 | 440 | 158 | 234 | 191 |
| Validation | 21 | 21 | 100 | 100 | 53 | 53 | 58 | 60 |
| Test | 21 | 21 | 100 | 100 | 53 | 53 | 105 | 98 |

Table 5: Random split of the four public small datasets for training, validation and test.

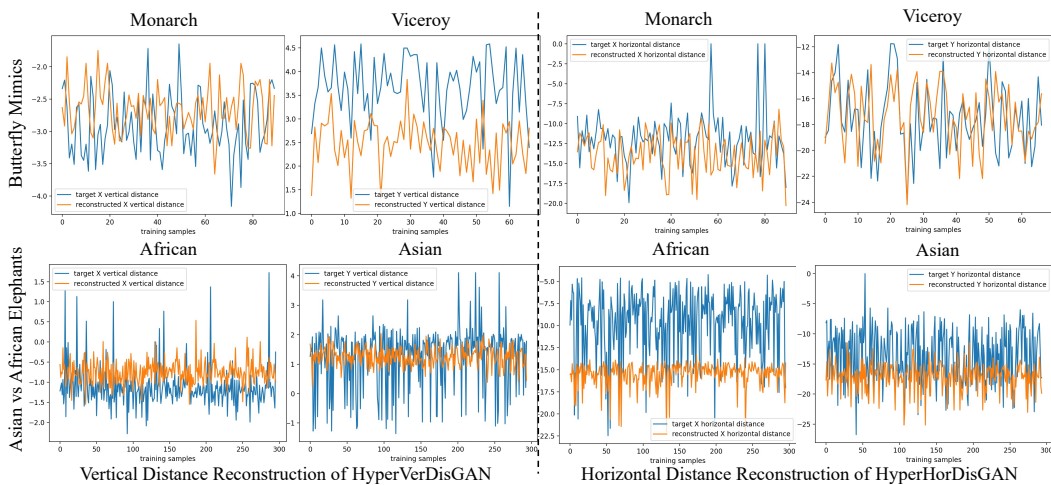

Figure 8: The curves of vertical and horizontal distance (Blue) and the reconstructed distance (Orange). The examples comes from an auxiliary classifier ConvNeXt (Liu et al., 2022).

## E.2 DATASETS SPLITTING

For all the experiments, four public small datasets are used: Butterfly Mimics, Asian vs African Elephants, mixed Breast Ultrasound and COVID-CT. Butterfly Mimics contains 132 monarch butterfly images and 109 Viceroy butterfly images. Asian vs African Elephants contains 494 asian elephant images and 496 african elephant images. mixed Breast Ultrasound contains 546 benign breast lesion images and 264 malignant breast lesion images. COVID-CT contains 307 non-COVID-19 images and 349 COVID-19 CT images. For each of the dataset, we randomly split them for training, validation and test. The details are listed in Table 5.

## F EFFECTIVENESS OF GENERATOR ON NATURAL DATASETS

We have explained the effectiveness of HyperDisGAN using two medical datesets in Section 5. We verified the distances reconstruction again using auxiliary classifier ConvNeXt as an example. The ability of the parameterized generator is shown in Figure 8 using two limited natural datasets.

## G TRAIN DATA DISTRIBUTION FOR STATE-OF-THE-ARTS CLASSIFIERS

We have shown the training data distribution for downstream ConvNeXt using two limited medical datasets in Section 4.3. Now we show training data distribution for ten state-of-the arts downstream classification models. Figure 9 compares the distribution of traditional augmentation (TA) + various auxiliary classifiers based HyperDisGANs on the two limited medical datasets by t-Distributed Stochastic Neighbor Embedding (t-SNE) (Van der Maaten & Hinton, 2008). The HyperDisGANs with ten different auxiliary classifiers tend to fill the hyperplane spaces of these ten downstream classification models, and generate samples along the margins of the hyperplanes.

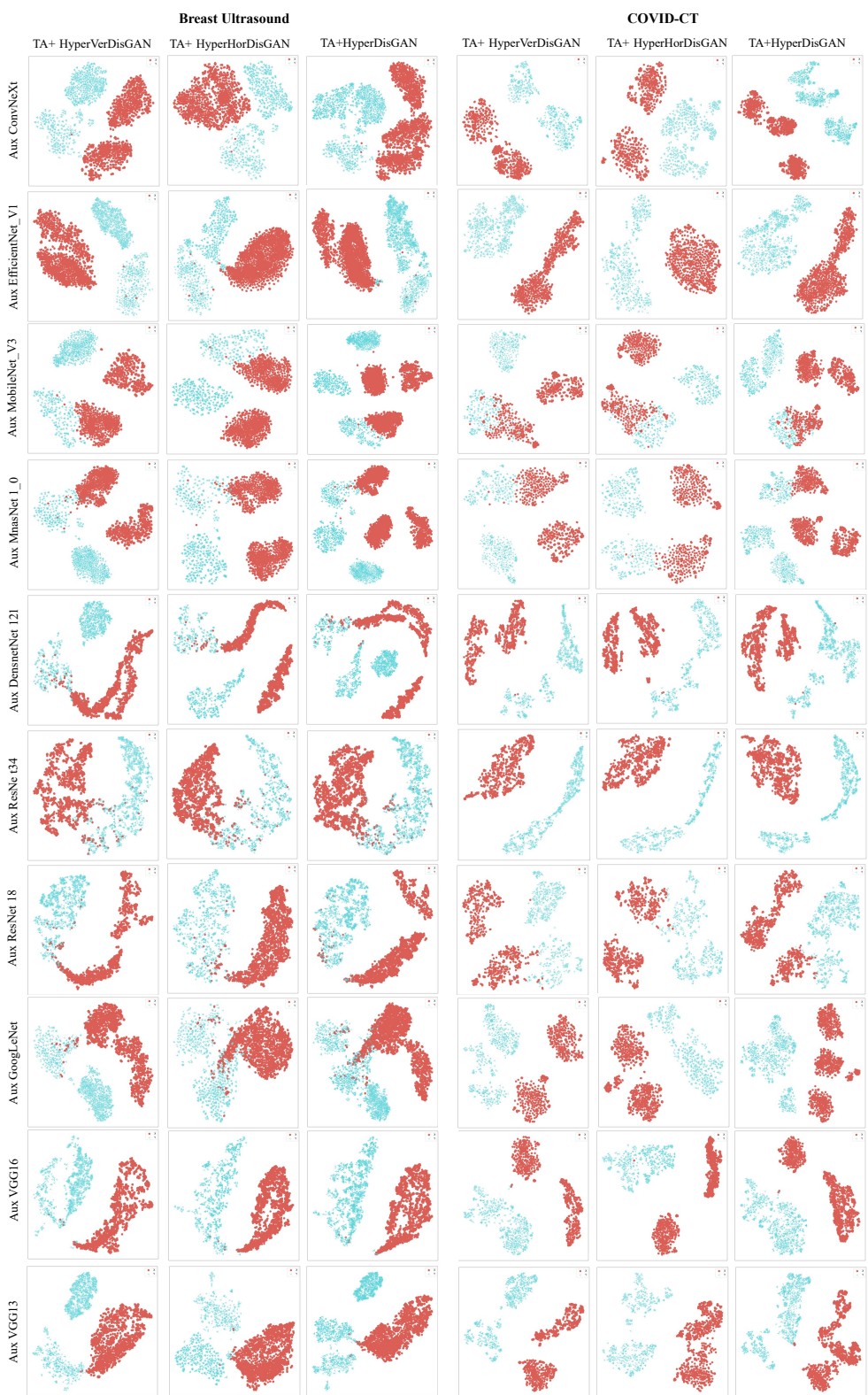

Figure 9: Training samples' distribution of the ten downstream classification models (Liu et al., 2022) visualized by t-SNE (Van der Maaten & Hinton, 2008). From up to low row with ten different auxiliary classifiers: ConvNeXt, EfficientNet V1, MobileNet V3, MnasNet 1_0, DenseNet 121, ResNet 34, ResNet 18, GoogLeNet, VGG16 and VGG13.

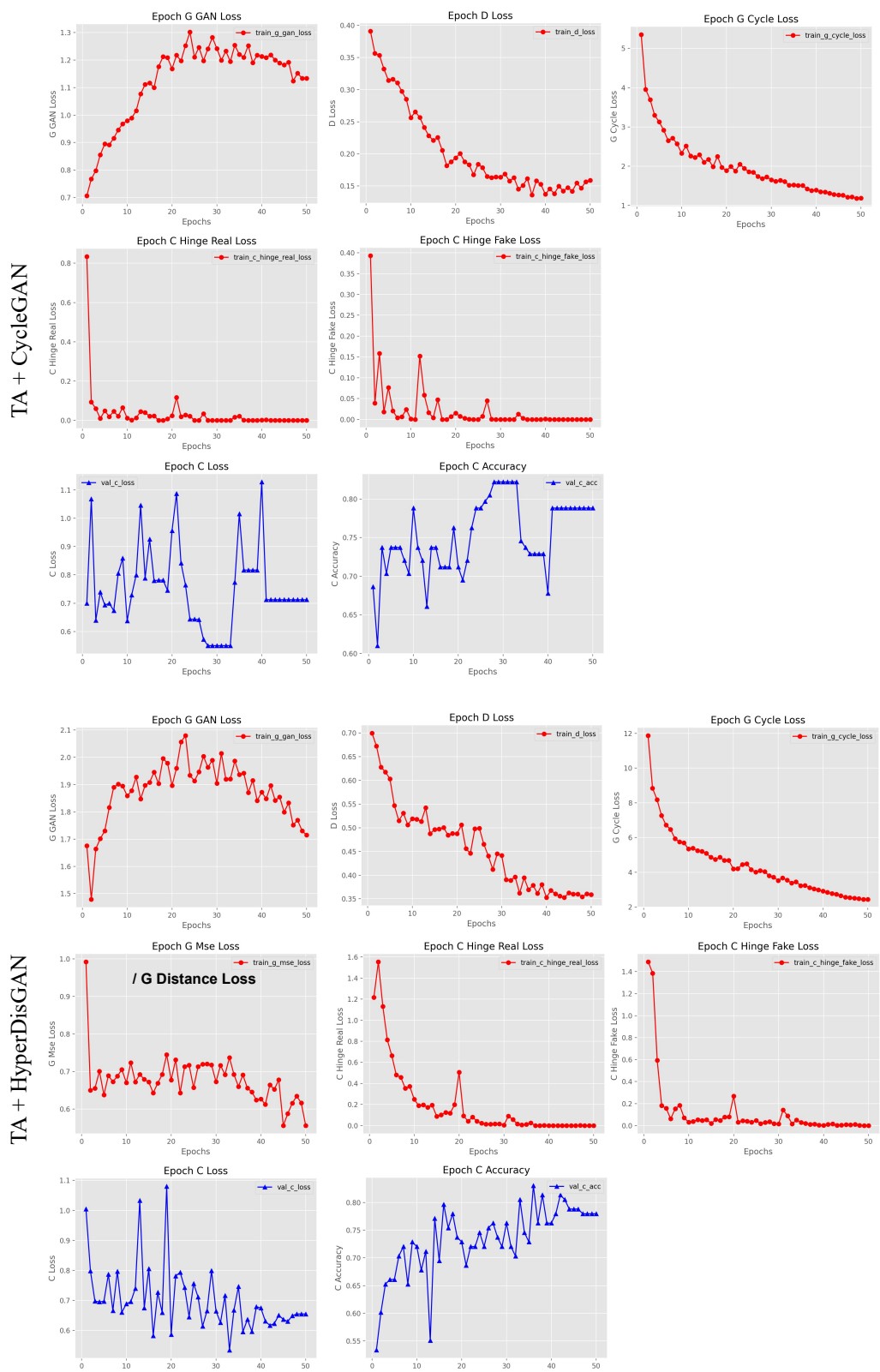

Figure 10: The traning loss curve (red) and validation loss curve (blue) for the traditional augmentation (TA)+CycleGAN and the TA+HyperDisGAN on COVID-CT Zhao et al. (2020) dataset. The example auxiliary classifier used in the HyperDisGAN is ResNet34.

## H  TRAINING AND VALIDATION LOSS CURVES

We show the curves of training and validation loss for the "traditional augmentation (TA)+CycleGAN" and the "TA+HyperDisGAN" respectively on the COVID-CT dataset. The loss of HyperDisGAN is the sum of the HyperVerDisGAN and the HyperHorDisGAN. As shown in Figure 10, the validation loss of the TA+HyperDisGAN gradually decreases even if in the later training phase, which is more stable than the TA+CycleGAN. Moreover, the distance loss of HyperDisGAN also gradually decreases, which further proofs the effectiveness of the parameterized generators.

## I  LIMITATIONS AND EXTENSIONS

The current work has limitations that need to be studied in the future: (1) The HyperDisGAN primarily focuses on the binary classifications, and exploiting hyperplane between multi-classes remains to be investigated. (2) Controlling the locations of generated samples utilizes the pre-trained auxiliary classifier constructing the hyperplane. We hypothesize that this can be an interactive process: the final binary classifier can be used again as the auxiliary classifier. This iterative process is expected to be the part where the artifical intelligence improves itself in the future. (3) A comprehensive evaluation of the generated samples in terms of usefulness beyond augmenting training samples.

### I.1  EXTENDING TO MULTI-CLASS CLASSIFICATION

The current method can be extended to multi-class classification, but we would like to start with the basical binary classification problem which is the foundation and starting point of machine learning. This extension work is in progress. Unlike the cross-entropy, multi-hinge loss solves the multi-class classification by constructing the multiple hyperplanes, this will be specifically divided into two ways: (1) OneVsRest: It takes a multi-class classification and turns into multiple binary classification for each class. (2) OneVsOne: It takes a multi class classification and turns into multiple binary classification where each class competes against every other class. Obviously, for $n$ classes, OneVsRest constructs $n$ hyperplanes whereas OneVsOne constructs $\frac{n(n-1)}{2}$ hyperplanes.

Upgrading the HyperDisGAN for multi-class classification is as follows: We firstly pretrain a multi-class classifier by the oneVsRest manner, and use $i^{th}$ hyperplane to classify between $i^{th}$ domain and not $i^{th}$ domain. Second, we can collect location information including the vertical distance between samples to hyperplane, and the horizontal distance between the intra-domain samples. Third, we can use only two generators (inter-domain and intra-domain) taking additional location information as input and one discriminator to control the generated samples' variation degrees by modifying the multi-domain generation method (such as the starGAN Choi et al. (2018)). The controllable generated samples will be used to adjust the decision boundaries for this multi-class classifier.

### I.2  LEVERAGING PRE-TRAINED GENERATORS

Restart training the generators to generate samples will costs resources and efforts. The proposed method is possible to be extended to leverage pre-trained generators, if the practitioners want to use their own dataset having same classes as our train dataset. The current input distance parameters representing the real target samples' location are not random enough. Now we should extend the generators to take any random distances as inputs. The range of the random distances can refer to the coordinate axis in Figure 7 and Figure 8. This is practical because we can still use the auxiliary classifier to reconstruct the random distances and impose the random distance loss. In the test phase, the practitioners can load the pretrained generators and take their own input images and any random distance to generate the large amount of samples.

## J  POTENTIAL NEGATIVE SOCIETAL IMPACTS

We believe the machine learning community should work together to minimize the potential negative societal impacts. Although the cross-domain generation benefits the downstream tasks, it should be used cautiously inspected for generating training data only. It shall not be used to tamper real patient data to jeopardize the life of patients.

