# OpenReview forum: "HyperDisGAN: A Controllable Variety Generative Model Via Hyperplane Distances for Downstream Classifications"
_ICLR.cc/2024/Conference — ICLR 2024 Conference Withdrawn Submission_

### Official Review · Reviewer_bUmq · 2023-10-23

**Soundness:** 2 fair
**Presentation:** 3 good
**Contribution:** 2 fair
**Rating:** 5
**Confidence:** 3

**Summary:**

This paper proposes a framework for data augmentation using GANs, inspired by CycleGANs. The main part of the method is to provide an explicit control between intra-class augmentation and cross-domain augmentation. This controllability is achieved by accessing the decision boundary of a pre-trained hinge classifier. The proposed data augmentation mechanism is then evaluated on several binary classification problems on image data.

**Strengths:**

* The proposed method is sound and reasonable.

* The experiments show that the method allows to improve the performance of a base classifier, and that it (often) outperforms other augmentation strategies.

**Weaknesses:**

* For now, the method is limited to binary classification. Could it be extended to multi-class classification? Self-supervised learning? Which are settings that benefit a lot from data augmentation.

* Experimental validation: I feel that the comparisons are not strong enough and would require a more careful and detailed evaluation. Notably, it is surprising that the Traditional augment often lowers the performance of the standard network, while data augmentation are most of the times beneficial in deep learning. It seems that the authors could have tried different settings and could have tuned better traditional data augmentations. For example, you test different hyper-parameters on your method. You should also test different hyper-parameters on the concurrent methods, otherwise it is natural that the max of HyperDisGan is higher than the max of the other methods.

* The proposed pipeline is a quite heavy pipeline with lots of hyper-parameters to tune.

* Clearness: the paper would benefit, at the beginning of Section 3, from a clear definition/formalization of the setting of the proposed method, e.g. input/output of the generator. This comes partly in the section 3.3, but it is not completely clear. For example, it is stated that the generator is a function from X (or Y) to X (or Y), but then, in the loss functions, the generator takes as input two variables, such as $G_{x2x}(x_1,-d_h(x_1,x_2))$. It is only in Section 4 that it is stated that the generator takes as input an image along with a distance variable replicated on the spatial dimension.

* Minor remarks on writing/typos: several use of "an" instead of "a", e.g. Figure 2 caption: "An data augmentation" or Section 3.2 "an pre-trained"; page 5 "intro-domain"; page 5: "transformating" -> "transforming"; Table 2: "HyerDisGan" -> "HyperDisGan".

**Questions:**

* What about latent transformations in pre-trained GANs? It has been shown that hyperplanes separate classes or modes in the latent space of a pre-trained unconditional GAN. Could you extend your method to leverage pre-trained generators? For practitioners who want to apply your method, it would be way less costly.

---

> ### Author Response · Authors · 2023-11-20
> **Response to Reviewer bUmq  (Part 1/2)**
>
> Dear reviewer bUmq,
> Thank you very much for your helpful comments! We respond to each of your comments one-by-one in what follows.
>
> **[W1] The limitation of the binary classification.**
>
> Good question, thank you. We add this topic in Appendix I.1 (Page 18) in new version paper.
>
> Binary classification is the foundation of classifications in Machine Learning. The current method can be extended to multi-class classification, but we would like to start with the basic binary classification problem which is the foundation and starting point of machine learning. This extension idea is straightforward. Unlike the cross-entropy, multi-hinge loss solves the multi-class classification by constructing the multiple hyperplanes, this will be specifically divided into two ways: (1) OneVsRest: It takes a multi-class classification and turns into multiple binary classification for each class. (2) OneVsOne: It takes a multi class classification and turns into multiple binary classification where each class competes against every other class. Obviously, for $n$ classes, OneVsRest constructs $n$ hyperplanes whereas OneVsOne constructs $\frac{n(n-1)}{2}$ hyperplanes.
>
> Upgrading the HyperDisGAN for multi-class classification can be as follows: we firstly pretrain a multi-class classifier by the oneVsRest manner, and use $i^{th}$ hyperplane to classify between $i^{th}$ domain and not $i^{th}$ domain. Second, we can collect location information including the vertical distance between samples to hyperplane, and the horizontal distance between the intra-domain samples. Third, we can use only two generators (inter-domain and intra-domain) taking additional location information as input and one discriminator to control the generated samples' variation degrees by modifying the multi-domain generation method (such as the starGAN [a] and the starGAN V2 [b]) conditioning on classes. The controllable generated samples will be used to adjust the decision boundaries for this multi-class classifier.
>
> Self-supervised learning benefit from the traditional data augmentation. For example, the augmentation in simCLR [c] directly flip/rotate the image and the class of the image is unknown. The proposed method cannot be used for self-supervised learning. Becasue training the HyperDisGAN needs to know the class label of each image in training dataset.
>
> [a] Choi, Y. et al. StarGAN: Unified Generative Adversarial Networks for Multi-Domain Image-to-Image Translation. CVPR 2018.
>
> [b] Choi, Y. et al. StarGAN v2: Diverse Image Synthesis for Multiple Domains. CVPR 2020.
>
> [c] Ting Chen. et al. A Simple Framework for Contrastive Learning of Visual Representations. ICML 2020.
>
> **[W2] Comparisons between the traditional and proposed method.**
>
> Thanks for your comments. Traditional augmentation (TA) sometimes cannot outperform the standard networks. That is the exactly the weak points we would like to raise and solve, i.e., relying on prior knowledge and specific scenes. For fairness of training on small-scale datasets, all experiments have been conducted on averaged over three runs. Furthermore, we test eleven classification models loading the ImageNet based pretraining-weights ("Training settings" in Page 6 in original version paper). Under this setting, making further breakthrough in classification performance is difficult even if the traditional augmentation is used. Therefore, we conduct the experiments combining the traditional augmentation and the proposed HyperDisGAN (TA+HyerDisGAN) to make breakthrough in classification performance, rather than aiming to replace the traditional augmentation methods.
>
> **[W3] Many hyper-parameters need to be tuned.**
>
> Thanks for your question. Actually, we don't need to tune all four hyper-parameters simultaneously. The four hyper-parameters are individual two paired parameters. The first pair $\lambda_{\rm verDIS}$ and $\lambda_{\rm crossCYC}$ is used to balance the loss of HyperVerDisGAN, whereas the second pair $\lambda_{\rm horDIS}$ and $\lambda_{\rm intraCYC}$ is used to balance the loss of HyperHorDisGAN. Therefore, we only need to tune the paired hyper-parameters for the HyperVerDisGAN and HyperHorDisGAN, respectively. Tuning these two versions of HyperDisGAN do not affect each other. See Appendix B, Table 3 (Page 14) in the new version paper.

---

> ### Author Response · Authors · 2023-11-20
> **Response to Reviewer bUmq (Part 2/2)**
>
> Coutinued.
>
> **[W4] Formulation clearness for proposed method setting.**
>
> Thank you for your suggestion. The $X$ and $Y$ in the Section 3.3 of original version paper only denotes two domains. For the convenience of reading, we have added a clear formulation at the beginning of Section 3 (Page 3) in the new version paper:
>
> Formulation:
>
> Our goal is to learn cross-domain mapping functions between $X$ and $Y$ and intra-domain mapping functions inside $X$ and $Y$, given training samples $\lbrace x\_i \rbrace_{i=1}^N$ where ${x_i}\in{X}$, and $\lbrace y_j  \rbrace_{j=1}^M$ where ${y_j}\in{Y}$. We denote the data distribution $x \sim P_X(x)$ and $y \sim P_Y(y)$, hyperplane vertical distance $d_x^v\in V_X$ and $d_y^v\in V_Y$ and hyperplane horizontal distance $d_x^h\in H_X$ and $d_y^h\in H_Y$. As illustrated in Figure 2, our model includes four mappings $G_{X2Y} : \lbrace X, V_Y \rbrace \rightarrow Y$, $G_{Y2X} : \lbrace Y, V_X \rbrace \rightarrow X$, $G_{X2X} :  \lbrace X, H_X \rbrace \rightarrow X$ and $G_{Y2Y} :  \lbrace Y, H_Y \rbrace \rightarrow Y$, where the generators generates images conditioned on both source image and target images' hyperplane distances. In addition, we introduce four adversarial discriminators $D_{X2Y}$, $D_{Y2X}$, $D_{X2X}$ and $D_{Y2Y}$, where $D_{X2Y}$ aims to discriminate between real images $\lbrace y \rbrace$ and generated images $\lbrace G_{X2Y}(x, d_y^v) \rbrace$; in the same way, $D_{Y2X}$ aims to discriminate between $\lbrace x \rbrace$ and $\lbrace G_{Y2X}(y, d_x^v) \rbrace$. $D_{X2X}$ aims to discriminate between real images $\lbrace x \rbrace$ and $\lbrace G_{X2X}(x, d_x^h) \rbrace$; in the same way,  $D_{Y2Y}$ aims to discriminate between $\lbrace y \rbrace$ and $\lbrace G_{Y2Y}(y, d_y^h) \rbrace$.
>
> **[W5] Typographical error.**
>
> Thank you, we have eliminate typographical errors in the new version paper.
>
> **[Q1] Pre-trained generators.**
>
> Good question, we have clarified this topic to Appendix I.2 (Page 18) in new version paper.
>
> Restart training the generators to generate samples will costs resources and efforts. The proposed method is possible to be extended to leverage pre-trained generators, if the practitioners want to use their own dataset having same classes as our train dataset. The current input distance parameters representing the real target samples' location are not random enough. Now we can extend the generators to take any random distances as inputs. The range of the random distances can refer to the coordinate axis in Figure 6 (Page 9) of original version paper. This is practical, because we can still use the auxiliary classifier to reconstruct the random distances and impose the random distance loss. In the test phase, the practitioners can load the pretrained generators and take their own input images and any random distance to generate the large amount of samples.

---

> > ### Comment · Reviewer_bUmq · 2023-11-20
> >
> > I appreciate the response from the authors. However, as my primary concerns remain unaddressed, and in alignment with the viewpoint of reviewer aNvk, I am adjusting my rating to 3: Reject. Below, I elaborate on specific points where I disagree with the authors:
> >
> > **Multi-label classification**: The expansion of the proposed method for multi-label classification tasks appears complex, with the authors presenting an idea without demonstrating its practical applicability. This is a major limitation of the proposed method.
> >
> > **Experimental rigor: hyper-parameter fine-tuning + comparisons**: The $\lambda$ values aren't the sole hyper-parameters; the choices regarding generator and discriminator architectures also qualify as crucial hyper-parameters. Additionally, my primary concern pertains to the apparent disproportion in the exploration of hyper-parameters between the proposed method and the image augmentation technique. Given the inherent instability of GANs and the meticulousness required for hyper-parameter fine-tuning in deep learning, it seems plausible that the proposed method might be more challenging to optimize than conventional augmentation methods. Furthermore, I echo reviewer aNvk's stance on the inadequacy of the experimental study, urging for more comprehensive comparisons with augmentation techniques like Mixup.
> >
> > **Use of pre-trained generators (question, not weakness)** : The authors did not get my question. When I mentioned pre-trained generators, I referred to pre-trained generators such as a pre-trained StyleGAN, i.e. a standard generator trained solely to generate images.

---

> > > ### Author Response · Authors · 2023-11-20
> > > **Response to Reviewer bUmq**
> > >
> > > Okay, thank you. As a matter of fact, we don't agree both you and the reviewer aNvk's rating. The reasons are as follows:
> > >
> > > The core innovation is solve the blinding generation of the GAN based models. Blinding generation making no contribution to adjusting decision boundary. But we find the connection between GAN based models and the downstream classifications: using the hyperplane theory to accuractly control the distances. We insist that our method reasonably solve the above problem.
> > >
> > > In brief, our experiments are not aim to replace the traditional augmentation methods with the GAN-based methods. We did't discard the generated samples of traditional methods in downstream classifications.
> > >
> > > **Multi-label classification**
> > > The expansion of the proposed method is easy to understand: control the multi-hyperplane distances for starGAN/starGAN V2. "We illustrate the expansion in detail and write a lot " did not means it is complex and unpratical. This paper is a starting point solving the weakness of GAN-based generation using binary classification. Multi-classification and its practical applicability is not our task in this paper.
> > >
> > > **Experimentak rigor: hyper-parameter fine-tuning + comparisons**
> > > The $\lambda$ values is corresponding to our two main innovations: vertical distance and horizontal distance, this is why we conduct experiments using the two pair lambda parameters. The selection of generator and discriminator architectures is not our subjects, we used the same architecture as the CycleGAN.
> > >
> > > Face of instability of GANs and challege of GAN's hyper-parameter fine-tuning, we have solved the label uncertainty by tuning two main distance parameters. GAN is famous for sample generation and augmentation. But it seems that the reviewers think that GAN-based generation is a joker compared with the traditional simple augmentation, and researching this joker is waste of time.
> > >
> > > Again, I recognize that mixup is simple and fast like other traditional augmentation methods. But, we aim to solve the weakness of GAN-based generation in augmentation, rather than replace the traditional methods with GAN-based models. Traditional methods also play a role of generating samples in our experiments.
> > >
> > > **Use of pre-trained generators (question, not weakness)**
> > > Our network is based on image-to-image translation rather than the noise-vector to image, so it cannot be extended to leverage pre-trained generators.

---

### Official Review · Reviewer_aNvk · 2023-10-25

**Soundness:** 2 fair
**Presentation:** 1 poor
**Contribution:** 1 poor
**Rating:** 1
**Confidence:** 5

**Summary:**

This paper studies data augmentation with generative adversarial networks (GANs). The paper proposes a Cycle-GAN based method, HyperDisGAN, which takes into account the hyperplane distance between and within classes to generate samples that are useful for training downstream classifiers. HyperDisGAN uses a classifier pre-trained by hinge loss to learn transformations so that the distance on the hyperplane is as large as possible for inter-class sample transformations, and so that the distance is small for intra-class sample transformations. Experiments were conducted mainly to compare the proposed method with the Cycle GAN baselines and confirmed that the proposed method slightly outperforms the baseline in accuracy and AUC.

**Strengths:**

+ The paper proposes an interesting data augmentation method based on CycleGAN that generates inter-class and intra-class samples by transforming real samples.

**Weaknesses:**

- The motivation to introduce the proposed method is weak. The issue presented by the paper is expected to be solved by a data augmentation method that interpolates between samples, such as mixup [a], but the paper introduces data augmentation by generative models without discussing this perspective at all.
- The proposed method is not practical. The proposed method introduces a pre-trained classifier $C_\text{aug}$, an inter-class generator $G_{x2y}$, and a intra-class generator $G_{x2x}$. These are not practical because they increase in proportion to the number of classes in the downstream classification task. In fact, the paper only evaluates the data set with a small number of classes.
- Despite the significant increase in computational complexity, the performance gain given by the proposed method is negligible.
- Experimental baseline is insufficient. Since the proposed method is a type of data augmentation, its performance should be evaluated by comparison with traditional/generative data augmentation methods. For example, traditional data augmentation methods such as mixup [a], CutMix [b], SnapMix [c], and generative data augmentation methods such as MetaGAN [d] and SiSTA [e] are appropriate as experimental baselines.
- Writing quality is poor. The paper contains many undefined words (e.g., "domain," "location," and "hyperplane"), which confuse the reader. In addition, the overall algorithm and procedure are not explained, making it difficult to grasp the overview of the proposed method. In general, the paper does not meet the quality required for an academic paper.

[a] Zhang, Hongyi, et al. "mixup: Beyond empirical risk minimization." International Conference on Learning Representations (2018).

[b] Yun, Sangdoo, et al. "Cutmix: Regularization strategy to train strong classifiers with localizable features." Proceedings of the IEEE/CVF international conference on computer vision. 2019.

[c] Huang, Shaoli, Xinchao Wang, and Dacheng Tao. "Snapmix: Semantically proportional mixing for augmenting fine-grained data." Proceedings of the AAAI Conference on Artificial Intelligence. Vol. 35. No. 2. 2021.

[d] Zhang, Ruixiang, et al. "Metagan: An adversarial approach to few-shot learning." Advances in neural information processing systems 31 (2018).

[e] Thopalli, Kowshik, et al. "Target-Aware Generative Augmentations for Single-Shot Adaptation." International Conference on Machine Learning (2023).

**Questions:**

Nothing to ask. Please see the weaknesses.

---

> ### Author Response · Authors · 2023-11-20
> **Response to Reviewer aNvk**
>
> Dear reviewer aNvk,
> We appreciate your efforts and detailed comments to improve the manuscript. Please find our responses below.
>
> **[W1] Interpolation based data augmentation.**
>
> Thanks for your comments. The motivation of our proposed method is not weak.
>
> Mixup does not bring high quality of synthesis and lack of explainability regarding the contributions of regions for the classification labels. Unlike this interpolation based methods, we build bridges between the image generation with the downstream classification, by accurately reshaping decision boundary for the downstream classification. The innovations based on hyperplane theory has good explainability. Our motivation is illustrated in the Figure 1 (Page 2) in original version paper, and we also upgrade the Figure 1 (Page 2) in the new version for better reading.
>
> Although mixup is a way of mixing classification, it brutally combines random pair of classes which ignore other possible generated point in the generation space unlike a GAN based method. Due to the limitation of the page and the limitation of the mix-up, we choose other generator-based data synthesis methods for comparisons.
>
> **[W2] Practicality of multi-class classification.**
>
> Thanks for your question. We add this topic in Appendix I.1 (Page 18) in new version paper.
>
> The proposed method is still practical for multi-class classification. The number of generators will not increase in proportion to the number of classes. We can take n-classes classification as the n binary classifications, and construct n hyperplanes by oneVsRest manner. For each hyperplane, we can still measure the vertical distance between samples to hyperplane, and the horizontal distance between the intra-domain samples. Moreover, starGAN [a] and starGAN V2 [b] only use one generator to transform images over the multiple domain. Therefore, what we need to do next is just let the starGAN's generator take additional distance parameters.
>
> [a] Choi, Y. et al. StarGAN: Unified Generative Adversarial Networks for Multi-Domain Image-to-Image Translation. CVPR 2018.
>
> [b] Choi, Y. et al. StarGAN v2: Diverse Image Synthesis for Multiple Domains. CVPR 2020.
>
> **[W3] Negligible gained performance.**
>
> All experiments are conducted on four datasets, including three small-scale datasets and one tiny-scale dataset. Small data is where we need data synthesis most. That's why we also test not only accuracy but also auc score averaged over three runs. Most auc scores increase by using our methods especially on the small-scale datasets. Last but most important, we test extensive classifiers including eleven different architectures. Therefore, the proposed method consistently improved the performance of classification.
>
> **[W4] Experimental baseline.**
>
> Thanks for your comments. We have selected cycleGAN which is published in ICCV conference, and ACGAN and VACGAN which are famous and highly relevant to auxiliary classifier. Of course in the scientific community, there are many other variants of GAN generating samples, but there are some page limit.
>
> **[W5] Undefined words and algorithm explanation.**
>
> We have FULLY explained these words in original version paper:
>
> The word cross-domain and intra-domain can be seen in the explanation of Figure 1 (Page 2): samples crossing the decision boundary are cross-domain, otherwise the samples are intra-domain; And in Related Work of ``Two-domain Image Transformation”, the related concept of domain is common in this research field.
>
> The word "location" can be firstly seen in line (from 9 to 11) of abstract: "The locations are respectively defined using the vertical distance ... and horizontal distance ..."; And in the line (from 6 to 8) of Page 2: "... the vertical distance ... and the horizontal distance ... are taken as the controllable location parameters".
>
> The word ``hyperplane" is in the first sentence of Section 3.1 (Page 3) : "We pretrain a auxiliary classifier by Hinge Loss (Rosasco et al., 2004) to obtain an optimal hyperplane ...". "hyperplane" is a common word in classification field of machine learning, such as SVM algorithm.
>
> We have explained the overall algorithm in both Figure 2(a) and Figure 2(b) in Page 4. Figure 2(a) denotes the training procedure of HyperDisGAN and Figure 2(b) illustrates joint training downstream classification models on the both original and HyperDisGAN's generated samples.
>
> Even though, we revised the full paper again to improve the quality. In new version, the overall algorithm and training procedure are detailed in Figure 2 (Page 4) and Figure 3 (Page 5).

---

### Author Response · Authors · 2023-11-20
**Response to Area Chairs**

Dear AC,
The reviewers misfocused the main contribution of the proposed method and therefore main points of our paper are not appreciated. Anyway, we do feel that a rating of 1 is not a fair rate for our paper. For example, from the Weakness 5 of aNvk, he/she reviewers cannot interpret hyperplane from hinge loss unless SVM is not a well-known concept, and misunderstood the whole idea of controlling the generated sample using the distance concept in the hyperplane space. Moreover, there are only two reviewers for this paper, not meeting the general case of ICLR review. Under this relatively unfair treatment, we withdraw our paper. Although we have some reservations about the reviewers’ opinions, we still thanks for AC and the reviewers’ efforts.

---

> ### Comment · Reviewer_bUmq · 2023-11-21
>
> Please let us try to be respectful. My review is based on the weighting of both strengths and weaknesses of the paper. I acknowledge your contribution and also highlight the main weaknesses of the proposed method. I do not believe that the contributions and potential impact outweigh the weaknesses. It doesn't look, at least to me, as an emotional review. Moreover, I do not have any "prejudice with GANs", since it is actually my main area of expertise.